# Exigency of Plant-Based Vaccine against COVID-19 Emergence as Pandemic Preparedness

**DOI:** 10.3390/vaccines11081347

**Published:** 2023-08-09

**Authors:** Anirudha Chattopadhyay, A. Abdul Kader Jailani, Bikash Mandal

**Affiliations:** 1Pulses Research Station, Sardarkrushinagar Dantiwada Agricultural University, Sardarkrushinagar 385506, India; anirudhbhu@sdau.edu.in; 2Department of Plant Pathology, North Florida Research and Education Center, University of Florida, Quincy, FL 32351, USA; 3Advanced Centre for Plant Virology, Division of Plant Pathology, Indian Agricultural Research Institute, New Delhi 110012, India

**Keywords:** biofarming, COVID-19, plant, SARS-CoV-2, transient expression, vaccines

## Abstract

After two years since the declaration of COVID-19 as a pandemic by the World Health Organization (WHO), more than six million deaths have occurred due to SARS-CoV-2, leading to an unprecedented disruption of the global economy. Fortunately, within a year, a wide range of vaccines, including pathogen-based inactivated and live-attenuated vaccines, replicating and non-replicating vector-based vaccines, nucleic acid (DNA and mRNA)-based vaccines, and protein-based subunit and virus-like particle (VLP)-based vaccines, have been developed to mitigate the severe impacts of the COVID-19 pandemic. These vaccines have proven highly effective in reducing the severity of illness and preventing deaths. However, the availability and supply of COVID-19 vaccines have become an issue due to the prioritization of vaccine distribution in most countries. Additionally, as the virus continues to mutate and spread, questions have arisen regarding the effectiveness of vaccines against new strains of SARS-CoV-2 that can evade host immunity. The urgent need for booster doses to enhance immunity has been recognized. The scarcity of “safe and effective” vaccines has exacerbated global inequalities in terms of vaccine coverage. The development of COVID-19 vaccines has fallen short of the expectations set forth in 2020 and 2021. Furthermore, the equitable distribution of vaccines at the global and national levels remains a challenge, particularly in developing countries. In such circumstances, the exigency of plant virus-based vaccines has become apparent as a means to overcome supply shortages through fast manufacturing processes and to enable quick and convenient distribution to millions of people without the reliance on a cold chain system. Moreover, plant virus-based vaccines have demonstrated both safety and efficacy in eliciting robust cellular immunogenicity against COVID-19 pathogens. This review aims to shed light on the advantages and disadvantages of different types of vaccines developed against SARS-CoV-2 and provide an update on the current status of plant-based vaccines in the fight against the COVID-19 pandemic.

## 1. Introduction

The sudden emergence of a deadly virus from Wuhan, China, at the end of 2019, and its subsequent rapid spread at pandemic speed throughout the world [1] forced us to lock down human society and halt our activities, leading to numerous horrific experiences. This virus is not only responsible for the loss of over 6.9 million lives [2], but it has also completely destroyed the world economy [3]. This has resulted in massive unemployment, food insecurity, and mental and physical stress among the human population, reminiscent of situations experienced during world wars. Furthermore, the WHO warns that this deadly virus “may never go away” and could become endemic in our global community. This persistent problem caused by the lethal virus is known as COVID-19, short for coronavirus disease 2019 (COVID-19). It is named after the pandemic outbreak of a novel coronavirus, initially named “2019-nCoV” and later renamed “SARS-CoV-2” by the Coronavirus Study Group (CSG) of the International Committee on Taxonomy of Viruses (ICTV) due to its significant genomic similarity with the severe acute respiratory syndrome coronavirus (SARS-CoV) [4]. On 30 January 2020, the World Health Organization (WHO) declared the COVID-19 outbreak a global public health emergency [5], aiming to minimize the threat in affected countries and reduce the risk of further international spread. However, to date, nearly 215 countries and territories and millions of people have been affected [2]. Therefore, the COVID-19 disaster is considered the biggest shock of the 21st century.

The ongoing threat posed by the coronavirus can only be curbed through the discovery and implementation of long-lasting solutions. The WHO has proposed numerous strategies, including source identification, individual and community transmission management, and global communication and collaboration as part of a global public health emergency response [6]. Above all, advanced research on therapeutics and vaccines is of paramount importance, as there are currently a limited number of clinically proven potential antiviral drugs or vaccines available against COVID-19 [7,8].Therefore, this pandemic necessitates the rapid development of drugs and vaccines to delay the spread of infection, thereby reducing the burden on hospitals and protecting the unexposed population.

A vaccine is a kind of formulation, mostly of microbial origin, that helps generate antibodies and stimulate immunity in warm-blooded animals against any diseases. Based on composition, these are different types, viz., pathogen-based vaccines [9], vector-based vaccines [10], nucleic acid-based vaccines [11], and protein-based vaccines [12].With growing concerns about the emergence of COVID-19 as a serious global health threat, there has been an international commitment to foster vaccine development to address these challenges [13]. While a range of vaccine strategies has been established, not all of them may be suitable or feasible in every case, especially during an epidemic. Although inactivated vaccines are the primary focus in the vaccine race [14], alternative approaches have been explored in parallel to open up new avenues in vaccine research. The production of vaccines during epidemic states should prioritize quality, scalability, rapidity, and costeffectiveness, which heavily depend on the availability of vaccine production platforms, including prokaryotic and diverse eukaryotic systems. In light of these considerations, researchers in various laboratories are exploring different platforms for vaccine development, viz., egg-based, and prokaryotic and eukaryotic cell-culture-based systems [15,16]. Among these, plant-based transient protein expression platforms, in combination with replicating and non-replicating viral vectors, have emerged as an excellent approach for producing peptide-based subunit vaccines and VLP-based vaccines [17]. Over the past decade, this plant-based platform has been favored over other platforms, including egg-based and cell-based systems, due to various advantages, including efficient production without contamination and the potential for direct oral delivery without the need for a cool chain system [18]. Furthermore, the transient expression of vaccines in plants is preferred due to the rapid production of large amounts of proteins with a low biosafety risk [19]. This plant-based transient expression platform has proven successful in generating viable vaccines against many human and animal pathogens, with several of them being considered potential candidates for evaluation in clinical trials and showing promising results. This underscores the importance of this technology in developing a vaccine against COVID-19. Thus, in this study, we explore the potential for developing a plant-virus-based vaccine to combat the pandemic surge of COVID-19.

## 2. Coronavirus Infection and Their Strain Variation

Like many other isometric viruses, Coronaviruses (CoVs) are single-stranded, positive-sense RNA viruses with spherical virion particles of 120–160 nm in diameter [20]. But they are named ‘corona’ due to their ‘crown-like appearance’ under the electron microscope (coronam; Latin term for crown) because of the presence of spike-like glycoprotein projections on their outer envelope measuring 20 nm in length [21]. The inner nucleocapsid core, which carries the 27–32 kb RNA genome (Figure 1), has a diameter of 9–11 nm [22]. Various avian and mammalian infecting coronaviruses are classified into four genera: alpha-coronavirus, beta-coronavirus, gamma-coronavirus, and delta-coronavirus, under the family Coronaviridae (subfamily Orthocoronavirinae), order Nidovirales [23]. They can infect a wide range of hosts, including birds and mammals. Alpha- and beta-coronaviruses (beta-CoV) typically infect mammals and can cause anything from the common cold to severe and fatal diseases, with respiratory and gastroenteritis symptoms [24]. On the other hand, gamma- and delta-coronaviruses primarily infect birds but can also infect other animals [25]. Avian coronaviruses often rapidly evolve and adapt to animals before making a species jump to humans [26]. In recent history, the severe outbreaks of Severe Acute Respiratory Syndrome (SARS), caused by the beta-CoV subgenusSarbecovirus (SARS-CoV-1) [27] and the Middle East Respiratory Syndrome (MERS), caused by the beta-CoV subgenus Merbecovirus (MERS-CoV) [28], highlight the potential dangers of coronaviruses. Both MERS and SARS have caused severe respiratory and intestinal ailments and have resulted in numerous deaths [29,30]. MERS, although more dangerous with a mortality rate of over 40%, spread more slowly and was mainly confined to the Arabian Peninsula, resulting in fewer than 100 deaths worldwide in 18 months [31]. In contrast, SARS was less fatal with a mortality rate of 10%, but it spread rapidly from person to person worldwide, infecting over 8000 cases within 8 months [31]. These outbreaks raise concerns about the potential emergence of future pandemics caused by coronaviruses.

Very recently, a similar outbreak occurred on a global scale, but in a more severe form, now known as COVID-19, which has shaken the world. It is caused by a novel coronavirus strain called SARS-CoV-2, which had not previously been identified in humans. So far, seven strains of human-infecting coronaviruses (HCoVs) have been identified [32]. Some of these strains belong to the genus Alpha-coronavirus (HCoV-229E, HCoV-NL63), while others, classified under the genus Beta-coronavirus, include milder strains like HCoV-OC43 and HCoV-HKU1, responsible for the common cold in immunocompetent individuals [33], as well as highly severe strains like MERS-CoV and SARS-CoV-1. SARS-CoV-2 is phylogenetically related to the previously reported SARS-CoV-1, with approximately 79% similarity, but it represents a new strain of the SARS virus (Figure 2). Like MERS-CoV and SARS-CoV-1, SARS-CoV-2 also originated from bats [34,35,36] and likely adapted to humans through an unconfirmed but probable intermediate mammalian source [37]. The zoonotic survival of SARS-CoV (civet cat) and MERS-CoV(dromedary camels) [38] and the higher transmission rate of the deadly SARS-CoV-2 strain pose a constant threat to public health [39].

The genome of SARS-CoV-2 is approximately 29.9 kb in size and encodes around 9860 amino acids from its 14 open reading frames (ORFs), which are further annotated into 27 proteins [40]. Interestingly, the exact size of the SARS-CoV-2 virus varies from 29.8–29.9 kb over time and space [41]. SARS-CoV-2 has a high mutation rate, which occurs randomly and can significantly alter its pathogenicity. Large-scale analysis of SARS-CoV-2 genomes has revealed clonal geodistribution and rich genetic variations within the population [42]. Initially, two lineages were reported to coexist: the S-type (earliest one) and the L-type, with the latter evolving from the former and being more prevalent in countries like the USA. The L-type spreads rapidly but is less virulent than the S-type [43].

Further analysis of thousands of complete genome sequences of SARS-CoV-2 isolated from human patients has demonstrated the evolution and genetic variation of the original strain of the novel coronavirus during its spread and human infections. Initially, the entire population was classified into three distinct population types, labeled A, B, and C [44]. Based on a comparison with a bat outgroup coronavirus, it was observed that the A-type is the ancestral type and closely related to the isolate BatCoVRaTG13 from Yunnan Province. However, the A-type is not predominant in East Asia; instead, its mutated versions are found in patients mainly from the USA and Australia. The B-types appear to have derived from the A-type through mutations (T8782C and C28144T) that lead to a functional change from leucine to serine in ORF8, and they have become prevalent across East Asia but are unable to spread outside this region without further mutations. The C-type, a distinct variant of its parental B-type through a G26144T mutation [44], is mainly distributed in Italy, France, England, and Sweden, and is considered the major European type [44]. These A, B, and C types have further been divided into finer subtypes, including A1a, A2, A2a, A3, A6, and A7 within the A-type; and B, B1, B2, and B4 within the B-type, based on RNA sequence analysis of 3636 SARS-CoV-2 isolates collected from 55 countries [45]. Among these subtypes, A2a is the most dominant and highly efficient in infecting humans. During its evolution from the ancestral O-type, the A2a type acquired an A23403G mutation in its RNA genome, resulting in an amino acid change from aspartic acid to glycine (D614G) located in the S1-S2 junction of the spike protein (S) [46]. This mutation may have an impact on the binding of the spike protein to the human angiotensin-converting enzyme 2 (ACE2) surface protein in the lungs, leading to subsequent entry into lung cells in larger numbers [45].

Immediately after the origin, the B-strain of the Wuhan population spread to different continents, especially Europe and the Americas, and rapidly changed to B.1 type gaining D614G mutation. This mutant mainly evolved in Europe and spread rapidly throughout the world, replacing the initial circulating strain (B type) [47]. The B.1 mutant underwent independent evolution that led to the emergence of different variants at different spatiotemporal scales. Similar to the influenza virus, the SARS-CoV-2 virus also mutates rapidly, and within just two to three years, researchers have identified multiple mutations (deletion and substitution) in the SARS-CoV-2 genome using ultra-deep sequencing [48], of which there are more than 30 mutations in the spike protein, especially within the receptor binding domain (RBD) [49]. Initially, B.1.1.7 lineage evolved in the United Kingdom during mid-2020, acquiring eight mutations (viz., Δ69–70 deletion, Δ144 deletion, N501Y, A570D, P681H, T716I, S982A, D1118H) in the spike (S) protein that led to its higher transmissibility, and virulence in Europe but also spread in all continents [50]. Furthermore, simultaneous evolution of two new lineages, viz., B.1.351 variants and B.1.1.28 (P.1) variants were recorded in South Africa in May 2020 and in Brazil in November 2020, respectively [51]. The B.1.351 variant received nine mutations (L18F, D80A, D215G, R246I, K417N, E484K, N501Y, D614G, and A701V) in the spike protein, of which three mutations (K417N, E484K, and N501Y) are located in the RBD and are mainly distributed in Europe [52,53]; whereas, B.1.1.28 variant harbors 11 mutations in the spike protein (L18F, T20N, P26S, D138Y, R190S, H655Y, T1027I, V1176, K417T, E484K, and N501Y) with three mutations (L18F, K417N, E484K) in the RBD and predominantly circulated in North and South America [53,54].Interestingly, these three variants possess N501Y substitution in the spike protein that shows an increased affinity of the spike protein to bind at ACE 2 receptors, thus enhancing the attachment to host surface, entry into host cells and infectivity of these variants as well as subsequent transmission [55].Later on, a deadly variant (B.1.617.2) was identified in India during December 2020 harboring ten mutations (T19R, (G142D*), 156del, 157del, R158G, L452R, T478K, D614G, P681R, D950N) in the spike protein [56]; of them, T478K, P681R, and L452R mutations made them highly infectious and immune to vaccination, spreading rapidly in many countries of the world [57]. But its predominance was reduced with the rapid emergence of a new B.1.1.529 variant and its lineages during early November 2021 in South Africa [58]. The multiple mutations in the nonstructural proteins and spike protein make these variants more infectious than previous counterparts. Thus multiple subvariants, like BA.1, BA.2, BA.3, BA.4, and BA.5, emerged at different times [59]. Different groups have assigned different names to these variants, but the Pango nomenclature system, proposed by [60] based on genomic epidemiology, provides important scientific information about the nature of changes in SARS-CoV-2 variants. To avoid confusion and complications in the public domain, the World Health Organization (WHO) has recommended the use of Greek letters (Alpha, Beta, Gamma, Delta) to name the variants [61], a nomenclature system that is followed worldwide. According to the WHO’s nomenclature system, the B.1.1.7 lineage is designated as the ‘Alpha’ variant, the B.1.351 lineage and its descendant lineages are called the ‘Beta’ variant, the P.1 lineage and its descendant lineages are assigned the name of the ‘Gamma’ variant, the B.1.617.2 lineage is called the ‘Delta’ variant, and the B.1.1.529 lineage and its descendant lineages are defined as the ‘Omicron’ variant [47,62]. Due to increasing transmissibility, reducing antibody neutralization ability, and severe pathogenicity, these variants were considered as variants of concern (VOCs), while numerous newly evolved strains, including Epsilon (B.1.427 and B.1.429), Zeta (P.2), Eta (B.1.525), Theta (P.3), Iota (B.1.526), Kappa (B.1.617.1), Lambda (C.37), and Mu (B.1.621), are regarded as variants of interest (VOIs) (Figure 2) because of the potential risk they may pose [47]. The emergence of new variants of SARS-CoV-2 is an ongoing process, and some of these variants may become more stable and infectious. Immune evasion by newly emerging strains is the major concern and challenge for public health [63,64]. Therefore, large-scale sequencing and monitoring of SARS-CoV-2 variants are necessary to understand the present status of mutations and genomic variations, which are crucial for antiviral drug and vaccine design.

## 3. Challenges in Vaccine Development during COVID-19 Outbreak

The goal of a vaccine is to trigger the immune response and provide protection against infectious pathogens. However, the efficacy and effectiveness of each product differ depending on the type of infection. Conventional vaccines were previously developed by attenuating or inactivating the respective pathogen, successfully reducing the spread and infection of pathogens and leading to the eradication of diseases like smallpox, polio, tetanus, diphtheria, etc. [65,66,67]. However, the production of such classical vaccines still has several flaws. Most importantly, the inactivated antigens are sometimes insufficient for inducing an immune response and may cause unexpected adverse reactions due to structural changes in the proteinaceous antigen or the presence of undesirable contaminants [68]. Additionally, insufficient inactivation or attenuation of the pathogen may still be harmful to individuals through reactivation [69]. Therefore, the expression of recombinant subunit vaccines and/or the production of whole-cell/virion vaccines are prioritized to avoid these limitations. These approaches involve targeted gene cloning and expression in different production platforms like *E. coli*-based bacterial cell culture, insect cell culture, mammalian cell culture, and plant-based systems.

The biggest scientific challenges for the production of billions of doses of safe and effective vaccines within a shorter time scale during a pandemic crisis rely on three important parameters. These parameters are the choice of a suitable scientific strategy to develop completely immunogenic vaccines, the availability of an efficient platform for mass vaccine production, and the rapid distribution of the vaccine to all parts of the world to ensure the complete eradication of virus reservoirs [70]. This initiative can only succeed if there is unprecedented cooperation between researchers, industries, and regulators. The primary goal of vaccine production is to develop the safest vaccines that are sufficiently effective in inducing an immune response in the long run. Since the vaccine targets for COVID-19 remain undefined until an outbreak occurs, the identification of epitopes for vaccine design is the foremost step [71]. However, developing a vaccine against SARS-CoV-2 is challenging due to the highly mutating nature of the virus. Similar to the influenza virus, the RNA genome of SARS-CoV-2 changes frequently [72]. Unlike HIV, SARS, and MERS, there is limited information available on the population dynamics of COVID-19[73]. Various genome-wide analyses of mutation mapping reveal an abundance of single nucleotide polymorphisms distributed within the entire genome [42,48]. Some of these mutations are hotspots for hypermutation and are located in essential viral genes. For example, the D614G mutation site in the spike protein [42]. Some nonsynonymous mutations result in changes in the amino acid sequence of the protein and may be directly linked to functional changes in viral pathogenicity. For instance, mutations in the spike protein may lead to changes in pathogenicity, such as the V367F mutation that enhances the protein’s affinity with the ACE2 receptor [74]. These mutations make COVID-19 more dangerous as they can enhance the virus’s ability to spread and cause antigenic drift, rendering existing antibodies ineffective in neutralizing the infection [75]. This strengthens the possibility of reinfection through the loss of immune memory. The tricky part is that the high mutation rate of COVID-19 may require the development of multiple vaccines, with scientists constantly coming up with new vaccine recipes to tackle the virus.

## 4. Developmental Status of Various Vaccines against COVID-19

A vaccine is the only option to render the novel coronavirus inertand safeguard human life. From various parts of the world, researchers are engaged in a rat race to find a vaccine, although it is a very lengthy and laborious process involving an array of steps that took more than a year since COVID-19 outbreak started. The key to the successful development of efficient vaccines is the design of an antigen and its delivery system to optimize antigen presentation, so that it can induce broad protective immune responses [76]. Moreover, massive global efforts are underway to develop potential vaccines, and antiviral drugs in order to slow down the spread of the COVID-19 and save lives. So far, about 90 candidate vaccines, including live-attenuated virus, whole inactivated virus, mDNA/mRNA, subunit vaccines, vectored vaccine, etc.,have been started against SARS-CoV-2 virus, each of which features specific advantages and limitations [77,78,79]. Of them, only 11 were approved by the WHO (Table 1).

### 4.1. Pathogen-Based Vaccines

The direct use of a pathogen as a killed or attenuated form is the most conventional method of vaccination. From ancient times, this strategy is successfully used for vaccination against the influenza virus, measles virus, and poliovirus. This strategy is now also being adopted by at least seven scientific groups for developing vaccines against COVID-19 [80]. The inactivated and live-attenuated/weakened form of a pathogen is utilized as an antigen for generating an immune response (Table 2; Figure 3).

#### 4.1.1. Inactivated Pathogen Vaccines

The inactivated vaccines contain the inactive (dead/killed) culture of pathogens, which are prepared by either heat/UV ray or chemical treatments (formaldehyde, β-propiolactone) [81]. These inactivated cultures of whole-celled pathogens completely lost the disease-producing ability, and when injected into a human body, it behaves as an antigenic molecule and activates an immune response to generate the necessary antibody. But, ironically in the killed vaccine, the conformational changes of the key epitopes present on the surface of the pathogen lead to poor immunogenicity [68]; thus, usually, multiple doses are delivered to achieve the desired response [82]. Despite that, this technology remains the primary choice for managing many pathogens, including Influenza, SARS, MERS, etc., since other technologies, such as recombinant polypeptides, have not been able to successfully imitate the pathogen’s epitope configuration. Thus, this classic strategy is efficiently used to produce vaccines for hepatitis A [83], flu viruses [84], and rabies [85].

Now, the same strategy is used to develop a vaccine for COVID-19. To develop a vaccine, various strains of SARS-CoV-2 were isolated from the bronchoalveolar lavage fluid (BALF) of patients from various geographical locations during the epidemic outbreak [86]. Only one predominant strain was purified and inactivated for vaccine development and other strains used as challenge strains during the preclinical trials.So far, a number of inactivated vaccines of SARS-CoV-2 have been developed; notably ‘Corona VAC’by Sinovac Life Sciences Co. Ltd. (China), ‘Sinopharm (BBIBP-CorV) vaccine’ by Sinopharm’s Beijing Institute of Biological Products, and ‘QazCovid-in’ by the Science Committee of the Ministry of Education and Science of the Republic of Kazakhstan, which were successfully released in the public domain [80]. The inactivated pathogen-based vaccines are easy to develop but have their own disadvantages. Usually, with the emergence of new strains, the immunogenicity of vaccines decreases, as evident against the ‘Omicron’ variant [80]. A booster dose of another vaccine (mRNAs, protein subunits) became necessary against the ‘Omicron’ variant even after injecting two doses of inactivated vaccine [87]. In spite of that, inactivated whole-celled pathogen vaccines are still a reliable option and are commercially released in many countries of the world, including China, India, Iran, Kazakhstan, and Turkey.

#### 4.1.2. Live-Attenuated Pathogen Vaccine

Live-attenuated vaccines (LAVs) contain the weakened or attenuated strains of pathogens created artificially by reducing their virulence through targeted modulation of pathogen-encoded interferon (IFN)-antagonists via genetic engineering [88], but still remaining viable. LAVs have historical success records against multiple human pathogens, including measles, mumps, rubella, chickenpox, etc. [89], and are the most frequently used for vaccination in humans to elicit strong cellular and humoral immune responses. Scientists from different parts of world were engaged in developing ‘Live-attenuatedSARS-CoV-2 vaccines’ using the latest codon deoptimization technology [88]. In addition, engineering of pathogenicity factors like deletion of polybasic furin cleavage sites in the spike protein [90], deletion of open reading frames (like ORF3 to ORF5) [91] or the inactivation of the nonstructural protein 16 (nsp16) 2′-O-methyltransferase [92] or amino acid substitution of the immune evasive viral protein deubiquitinase [93] leads to genetation of the attenuated strain of coronaviruses with immunogenic potential. These technologies are encouraging for the development of LAVs for prophylactic, active immunization against coronavirus in humans and are expected to provide long-lasting protection with an anticipated safety profile similar to other licensed vaccines for active immunization [94]. One such LAV (COVI-VAC) is formulated by Codagenix in collaboration with the Serum Institute of India and its single dose intranasal delivery showed a promising response in Phase III of clinical trials [95]. The other reports of LAVs also have shown their potential against SARS-CoV-2 in animal models [90,96,97]. Although the development process of LAVs are lengthy and time-consuming, their ability to induce strong cellular and humoralimmune responses make them suitable for durable immunization against SARS-CoV-2 [98]. Furthermore, indirect dispersion of a live-attenuated virus from vaccinated individuals to their contacts contributes in reaching rapid herd immunity in the population [99]. In spite of several attractive advantages, some challenges still persist in their mass application. Primarily, the failures of LAVs are ascribed due to the mutation in the surface antigens of pathogens. In addition, the frequent recombination among SARS-CoV-2 strains [100] increases the chance of LAV virulence reversion [101] as exemplified previously in Dengue virus [102]. Therefore, extensive screening and monitoring are necessary to identify the risk of ‘virulence reversion’ and their transmission into the community.

### 4.2. Recombinant Vector Vaccines

To avoid the risks associated with live-attenuated vaccines, especially for HIV and malaria, the search for a safer alternative has led to the development of gene expression vectors using a virus genome [103]. Infectious, benign viruses are genetically engineered to design a gene-delivery vehicle. So far, various viruses, such as adenovirus (Ad), adeno-associated virus (AAV), lentivirus, and vesicular stomatitis virus (VSV), have been exploited for the construction of suitable vector systems and evaluated for the expression of various proteins that can be easily introduced into human cells [75,104]. Such vector constructs can be used to integrate the structural genes, including spike (S), N, matrix (M), and envelope (E), of SARS-CoV and express them alone or in combination in the human body as specific antigens [105]. These genetically attenuated or weakened virus-vectors induce an immune response through the expression of the antigens rather than causing disease. These vectors can be either replicating or non-replicating types (Table 2; Figure 3) and are now targeted for generating vaccines against COVID-19.

#### 4.2.1. Replicating Viral Vector-Based Vaccine

In this case, the foreign gene is inserted into the competent viral vector without replacing any vector sequences; thus the virus vector remains functional in terms of replication and movement. Viruses like adenovirus, cytomegalovirus (CMV), measles virus, and vesicular stomatitis virus (VSV) are engineered into the replicating vector form [106]. Usually, a low dose is sufficient to generate an immune response [107]. Typically, avirulent strains of the virus are chosen for vector construction, leading to harmless infection in immune-competent hosts. However, preexisting immunity against the virus vector in the human body results in reduced replicability of the construct [108], thus generating very poor immunity to the antigen encoded by the foreign genes. Moreover, the limited cargo capacity of the replicating vector limits their utility in carrying larger antigens [109]. Nevertheless, the possibility of the evolution of a mild and ‘engineered’ virus vector into its wild form increases the risk to human safety [103]. Thus, considering both safety and immunogenicity issues, this category of vaccines is less preferred for human application. Despite that, various replication-competent chimeric virus/recombinant virus vectors carrying the Spike (S) glycoprotein have been developed by different groups, such as Brilife (developed by Israel Institute for Biological Research), AdCLD-CoV19 (developed by Cellid Co. Ltd., Seoul, Republic of Korea), AVCOVID-19 (developed by Aivita Biomedical, Inc., Irvine, CA, USA), [110] and are still in the clinical trial stages.

#### 4.2.2. Non-Replicating Viral Vectors

To avoid the risk of infectivity, replication-incompetent vectors are designed using various virus genomes, including adenoviruses, adeno-associated virus, alphavirus, and herpesvirus. In these cases, the structural genes are deleted from the replicon to accommodate a large foreign gene [106]. Although unable to replicate, these replication-defective forms are physically and genetically very stable with a large cargo capacity to carry one or more foreign genes [111]. They are also employed to efficiently express vaccines. Moreover, the chance of reversion of pathogenicity is minimal [109], making them very safe to use. Currently, many scientific groups have developed vaccines against SARS-CoV-2 using various non-replicating viral vectors such as recombinant adenovirus (AdV), modified vaccinia Ankara (MVA) virus, etc. [112,113]. Among them, different adenovirus-based non-replicating vectors, such as human adenovirus (Ad26 and Ad5) and chimpanzee adenovirus (ChAdY25), were most commonly employed for vaccine design [114]. Examples include vaccines developed by Janssen Pharmaceuticals (Ad26.COV2.S), the University of Oxford in collaboration with AstraZeneca (ChAdOX1-nCoV), and Gamaleya Research Institute Russia (Gam-COVID-Vac/Sputnik V), which express the spike protein of SARS-CoV-2 [115]. These vaccines have been approved by the WHO and authorized for several countries due to their strong immunological potentiality in clinical trials. However, the association of these vaccines with thrombosis (blood clotting) and thrombocytopenia (i.e., low platelet counts) syndrome has been reported in some postauthorization surveillance studies [116]; thus, further studies are necessary to reduce the risk associated with these groups of vaccines.

### 4.3. Nucleic Acid-Based Vaccines

To combat COVID-19 with a single-dose vaccine, nucleic acid-based vaccines are preferred by various scientists as attractive alternatives to direct pathogen vaccines or virus vector-based vaccine candidates, due to the biosafety risk and poor immunogenicity issues associated with the latter. The genetic information (DNA or RNA) in nucleic acid-based vaccines encodes the protein (antigen) inside human cells once inserted, inducing an immune response [11]. The spike protein of SARS-CoV-2 is predominantly delivered for designing these vaccines [117]. Looking forward to innovation, RNA and DNA vaccines have the potential to develop much more quickly due to the rapid construction of ‘purified’ synthetic nucleic acids, making them easier to scale up in large volumes compared to traditional vaccines (Table 2; Figure 3).

#### 4.3.1. DNA Vaccine

DNA vaccines consist of plasmid DNA constructs carrying the codon-optimized gene sequence of SARS-CoV-2. The DNA construct is designed by incorporating the gene encoding proteins of the pathogen within an improved plasmid DNA vector associated with donor and acceptor splice sites, heterologous viral RNA exporter, and post-transcriptional regulator elements [118]. Once administered, these DNA sequences start encoding proteins to elicit superior cellular and humoral immune responses. Between immunization with a DNA template and expression of the target antigen, the DNA has to overcome the cytoplasmic membrane and nuclear membrane, be transcribed into mRNA, and move back into the cytoplasm to initiate translation [119]. DNA vaccines have been characterized by suboptimal potency due to low transfection and lesser protein expression in clinical trials [120]. Furthermore, the challenge associated with the delivery of DNA vaccines into the cell nucleus is the major reason for their relatively low immunogenicity profiles. Enhanced delivery technologies, such as electroporation, have increased the efficacy of DNA vaccines in humans. However, a caveat of DNA vaccines is the potential risk of integration of exogenous DNA into the host genome via homologous recombination [121], which may cause severe mutagenesis and induce new diseases. Despite that, various scientific groups are directly engaged in the design of DNA vaccines against COVID-19 using different gene delivery strategies [122,123,124]. Several genes encoding the S, N, M, and E proteins of SARS-CoV have been tested in mice for their efficacy [125]; among them, DNA vaccines encoding the spike (S) glycoprotein are most common [126]. So far, a few DNA vaccines, such as INO-4800 (Inovio Pharmaceuticals, San Diego, CA, USA), GX-19 (Genexine, Inc., Seoul, Republic of Korea), AG0301-COVID19/AG0302-COVID19 (AnGes, Inc., Osaka, Japan), bacTRL-Spike (Symvivo Corporation, Collingwood North, Australia), and Covigenix VAX-001 (Entos Pharmaceuticals Inc., Alberta, Canada) have been developed with the aim of monoclonal antibody production in vivo via DNA plasmid expressing spike protein into a patient [122]. Among them, ZyCoV-D (the first DNA vaccine for COVID-19) produced by Zydus Cadila Healthcare Limited, India, is approved for emergency use in India [127], and the rest have already reached different phases of clinical trials, attesting to their safety as vaccines against SARS-CoV-2. However, most of these vaccines were discontinued after phase II/III of clinical trials, possibly due to their low immunogenic ability.

#### 4.3.2. RNA Vaccine

Currently, two forms of mRNA vaccines are available, including conventional mRNA vaccines and self-amplifying mRNA vaccines, derived from the positive-stranded RNA genome of the targeted viruses [119]. These vaccines are mostly delivered into host cells through lipid nanoparticles (LNPs) [128], although other lipid-based delivery systems such as lipoplexes and polyplexes are also available [128]. Self-amplifying mRNA vaccines have the ability to replicate themselves through the synthesis of the RNA-dependent RNA polymerase complex, generating multiple copies of the antigen-encoding mRNA and highly expressing heterologous foreign proteins that mimic the antigens in vivo [119]. Compared to the rapid expression of conventional mRNA vaccines, self-amplifying mRNA vaccines induce both humoral and cellular immune responses more slowly but confer equivalent protection at a much lower RNA dose [129,130,131]. The replicon in these vaccines lacks viral structural proteins and does not produce infectious viral particles. Furthermore, both conventional mRNA and self-amplifying mRNAs cannot integrate into the host genome and naturally degrade during the process of antigen expression [132]. To date, approximately five mRNA vaccines are in clinical trials, and two RNA vaccines, mRNA-1273 (SPIKEVAX) made by ModernaTX, Inc. (Cambridge, MA, USA) and BNT162b2 (COMIRNATY^®^) made by Pfizer BioNTech (New York City, NY, USA) have been approved by WHO and successfully released commercially [133]. However, some reports have raised concerns about different side effects, such as cardiac arrest, due to myocarditis or pericarditis shortly after mRNA vaccination [134,135], questioning the safety of mRNA vaccine administration.

### 4.4. Protein-Based Vaccines

To avoid the health risks associated with the use of whole pathogens or their nucleic acids as vaccines, many scientists advocate for the direct use of antigenic proteins derived from the pathogens [136]. Designing these vaccines is relatively easy and safe but requires proper purification before use. These antigenic proteins can be produced as individual protein subunits or as well-assembled particle forms that mimic the structure of virus particles (Table 2; Figure 3). Both forms of vaccines have significant utility.

#### 4.4.1. Recombinant Protein Subunit

Protein subunits composed of one or more different types of viral antigens are used as vaccines against a large number of pathogens. To produce such antigens, the targeted gene of the pathogen/virus is separately cloned into a suitable expression vector using recombinant DNA technology, and their heterologous expression is performed using various prokaryotic (e.g., *E. coli*) and eukaryotic expression systems (e.g., yeast, insect cells, mammalian cells, plant cells) [137]. Compared to other types of vaccines, the production of recombinant protein subunit-based vaccines is more secure and can be easily scaled up in a cost-effective manner [138]. When administered into the human body, these vaccines often require suitable adjuvants (such as aluminum hydroxide gel) to elicit a strong immune response. Subunit vaccine formulations are prepared by mixing specific purified antigens with potent adjuvants [139]. The development of subunit vaccines through the expression of the spike protein of SARS-CoV using a baculovirus-based expression system, and their successful evaluation in mice [140], served as the basis for accelerated development of subunit vaccines against COVID-19 [141]. Many recombinant subunit vaccines against COVID-19 are currently in the clinical trial stage, and most of them express the spike (S) glycoprotein of SARS-CoV-2 through different expression platforms [6]. Among them, NVX-CoV2373 (Nuvaxovid and Covovax) is approved for use in more than 40 countries or regions, mostly as primary doses [12], sometimes as a booster dose [142]. The ability of such subunit protein-based vaccines to elicit an immune response in humans is well recognized, but close surveillance is necessary to identify any adverse effects in the vaccinated population.

#### 4.4.2. VLP-Based Vaccines

Although stable and safe, sometimes subunit vaccines are not sufficient to induce adequate long-term immunity, requiring higher multiple doses that make them very expensive. These subunit proteins can be made more efficient by self-assembling into particle forms that mimic the architecture of natural antigen molecules like viruses, known as VLPs [143]. VLPs are multiprotein structures without the incorporation of a viral genome [144] and are capable of displaying several antigens to design multi-epitopic vaccines [145]. Different platforms, such as prokaryotic and eukaryotic expression systems, are used for the production of VLPs, and further purification and characterization are necessary to use them as vaccines. Once vaccinated in the human body, VLPs mimic natural virus infections, stimulating both strong cellular and humoral responses [146]. The development of VLPs or nanostructures displaying coronavirus antigens has multiple advantages [147], as they are structurally analogous to coronavirus particles and are easily and efficiently recognized by antigen-presenting cells due to their similar size, triggering the adaptive immune system [146]. Previously designed VLPs expressing SARS-CoV-1 and MERS antigens provided important guidance for the development of VLP-based vaccines against COVID-19 [148,149]. Various attractive platforms like SpyTag/SpyCatcher technology, Proficia^®^, Medicago’s plant-based platform, are used for expressing VLP vaccines in different host systems such as bacteria, fungi (yeast), insects, mammals, and plants [150]. A VLP-based vaccine developed by SpyBiotech-Serum Institute (Pune, India) is composed of a VLP part using HbsAg (surface antigen of HBV) exposing the receptor-binding domain (RBD) of the spike protein S1 subunit [151], expressed in yeast cells. Similarly, two monovalent eVLP-based vaccines (VBI-2902 and VBI-2905) and one multivalent eVLP-based vaccine (VBI-2901) expressing spike proteins of different coronaviruses, namely SARS-CoV-2, SARS-CoV and MERS-CoV on the VLP’s surface were developed by VBI Vaccines Inc. (Cambridge, MA, USA) [152], and another VLP expressing all four structural proteins (S, E, M, and N) of SARS-CoV-2 was produced by the Scientific and Technological Research Council of Turkey [153]. Subsequently, many other VLP-based vaccines were developed and placed in clinical trials [154]. However, no VLP-based vaccines against COVID-19 have received approval yet. Most of the VLP-based vaccines were produced in different expression systems such as *E. coli*, fungi, or insect cells, each with its own pros and cons.

## 5. Why Is a Plant-Based Platform Preferred for Vaccine Production against COVID-19?

Once an effective vaccine has been identified, it will need to be rapidly produced on a massive scale, potentially requiring billions of doses. Therefore, the selection of an appropriate platform is crucial. Various platform technologies have been explored, ranging from egg-based systems to cell-based systems, prokaryotic to eukaryotic systems, each with its own advantages and disadvantages [137]. The choice of the expression platform is not only a matter of convenience for investigators but also essential for the vaccine’s efficacy inside the human body. To ensure proper functioning of complex therapeutic proteins, they must be processed and folded correctly to achieve the desired biological activity [155]. It is necessary to select a eukaryotic system that incorporates humanized glycosylation and phosphorylation pathways [156]. While mammalian cell systems are considered the best option, the risk of contamination with animal/human pathogens poses challenges in quality assurance [157]. To address these issues, plant-based platforms can be promoted as the preferred alternative to eukaryotic mammalian systems.

Plant-based vaccines are the kind of recombinant vaccines that are produced in selected plants. Plant-based vaccines can be produced through either transgenic or transient approach (Figure 4). In the transgenic method, the manufacturing of vaccines primarily entails the expression of antigens into plant cells through nuclear or plastid transformation of the transgene(s) encoding the antigen [158]. Commonly, the stable transformation leads to the lower expression of subunit antigens in the transgenic plant cells, ranging from 0.01 to 0.30% of total soluble plant protein [159], whereas the transient expression involves the production of desired protein or antigen soon after the agrobacterium-based delivery of heterologous gene in the host cells via injection or vacuum infiltration [160]. Both binary and plant virus-based vectors system like magnICON are used to achieve high-level expression of foreign genes in the transient strategy. Usually, a higher yield of protein/antigens can be achieved rapidly within shortest time span [161]. There is no need of stable transformation; thus plant regeneration using tissue culture is not necessary. Thus, transient expression system is being commercially exploited to produce a number of medically important vaccines and biologics.

Plant-based platforms are preferred primarily because they are highly cost effective and very easy to grow and do not require aseptic environmental conditions or expensive growth media, as is the case with bacterial and mammalian cell culture-based vaccine production platforms [162]. Plant-based vaccine production has the potential to be cost effective compared to other platforms. The lower cost is attributed to several factors, including the use of inexpensive plant systems, reduced infrastructure requirements, and simplified downstream processing [163]. Plant-based production eliminates the need for expensive cell culture systems or specialized fermentation equipment, reducing overall production costs. Plants can be grown in large quantities in the field or in controlled environments, leading to potentially lower manufacturing costs compared to other systems [164]. Additionally, plant-based platforms do not harbor human pathogens, minimizing the risk of contamination or transmission of toxic/pathogenic compounds during vaccine production and processing [165]. In some cases, no special processing is required if vaccines can be generated in edible plant parts [166], making them suitable for oral delivery. Furthermore, plant cells have the ability to modify target proteins in eukaryotic ways, including N-linked glycosylation, that are strikingly similar to those found in mammalian cells [167]. Furthermore, plant systems allow for the production of properly folded complex proteins (vaccines) that can elicit both cellular and humoral immune responses in humans [168]. When delivered as an oral vaccine, they can stimulate the mucosal defense in the mouth and nose, preventing the entry of infectious virus particles like SARS-CoV-2. Thus, plant-based expression systems can be preferred as a cost-effective alternative for eukaryotic vaccine production.

Another major challenge in producing large quantities of vaccines is scaling up the manufacturing process quickly. The infrastructure required will vary depending on the vaccine type and platform selected. Vaccines must be produced under good manufacturing practice (GMP) conditions to save time and costs [169]. In the recent COVID-19 pandemic, time remains a major hurdle for effective vaccine development, highlighting the need for approaches that allow extremely fast development and large-scale production with rapid licensing to prevent global outbreaks. The cost associated with vaccine production is also a limiting factor. Cultivating whole plants is easier than using cell suspension cultures, and it is easier to scale up for manufacturing purposes [170]. Plant-based vaccine production can offer faster production times compared to traditional systems. This is mainly due to the ability to rapidly scale up production using transient expression in plants like *Nicotiana benthamiana* [171]. Plant-based systems can generate high yields of recombinant proteins within a matter of weeks, allowing for more rapid vaccine production compared to other platforms [171]. Thus, scalability of plant-based systems makes them particularly suitable for emergency situations or rapid response to emerging infectious diseases. This allows for the production of billions of vaccine doses within a limited time frame. Transient expression systems in plants make it easier to produce subunit proteins and VLPs in whole plant tissue, ensuring rapid protein yield without requiring tedious regeneration processes and avoiding the drawbacks of stable transformation [172]. This strategy is highly cost effective and provides satisfactory yields. Comparative analysis of recombinant subunit protein production in different platforms illustrates the potential benefits of plant-based systems over traditional expression platforms such as bacterial, insect, and mammalian cells [173,174]. Establishing a large-scale production system with lower initial capital investment and maintenance costs compared to traditional platforms provides unique advantages in favor of plants [174]. Moreover, the higher productivity and better yield per unit biomass, along with the higher plant biomass per hectare per year, ensure higher profit margins. Additionally, the rapid upscaling of production in a shorter time span through transient expression in plants offers further advantages [175,176,177]. This enables the affordable production of strain-specific vaccines in a timely manner, serving as a rapid-responsive strategy against quickly evolving pathogens.

Recombinant protein purification from plant-based expression systems poses unique challenges, particularly when aiming to produce plant-based vaccines and therapeutics. To address this issue, researchers have explored the use of affinity tags to facilitate facile protein purification. Affinity tags, such as His-tag, GST-tag, and Strep-tag, are short peptide sequences genetically fused to the target recombinant protein. These tags enable selective and efficient purification using specific ligands or matrices. Numerous studies have successfully demonstrated the application of affinity tags for recombinant protein purification in plants. For instance, the use of hydrophobin fusions in *Nicotiana benthamiana* has allowed high-level transient protein expression and purification [178]. Similarly, the rapid production of SARS-CoV-2 receptor binding domain (RBD) and spike-specific monoclonal antibodies in *Nicotiana benthamiana* using affinity tags has shown promise for plant-based COVID-19 vaccine development [179,180]. Moreover, self-replicating viral vectors linked to flagellin have been employed to enable efficient purification of the receptor-binding domain of SARS-CoV-2 [181]. These examples collectively highlight the feasibility of using affinity tags to overcome protein purification challenges in plant-based systems. These findings underscore the potential of affinity tags for the development of plant-based vaccines and therapeutics, including those against COVID-19.

Global distribution of vaccines is another significant challenge, particularly in reaching remote areas and eradicating the virus from its last reservoirs during a pandemic. Traditional prokaryotic and eukaryotic platforms require a sound supply system aided by a cold chain [182]. Cold chain requirements pose challenges for vaccine distribution, particularly in resource-limited settings or areas with limited access to refrigeration facilities. Plant-based vaccines may have specific cold chain requirements to maintain stability and potency. Some plant-derived vaccines may require storage and transportation at refrigerated temperatures. To address these challenges, efforts are underway to develop plant-based vaccines with improved thermostability [183]. This involves engineering the vaccine antigens or incorporating stabilizing agents to enhance resistance to temperature variations. Another approach is the development of alternative delivery formats, such as lyophilized or tablet formulations, which can improve the stability and reduce the dependence on cold chain requirements [184]. Similarly, the use of plant-based vaccine platforms for oral delivery also offers advantages, as it bypasses the need for cold chain storage and distribution. Sometimes, plant-based vaccines remain stable for extended periods without the need for a cold chain, such as in the form of dry seeds or edible leaves/fruits. They can serve as ideal booster vaccines, eliminating the requirement for multiple doses of traditional vaccines [160]. This not only reduces vaccine costs but also ensures uniform allocation at economically affordable prices in the developing world [185,186]. During times of urgency, researchers, regulatory bodies, and manufacturing industries need to collaborate to meet the peak demand. The industry should facilitate good manufacturing platforms suitable for large-scale vaccine production in parallel with research and development of vaccines targeting the latest virus strains, while regulators should reduce the time required for clinical testing to support global vaccination efforts. In such situations, employing plant-based platforms would be the best choice to address these challenges and to be prepared for future potential pandemic threats (Table 3).

Regarding the safety of plant-based vaccines, studies have demonstrated their favorable safety profiles. For example, Margolin et al. [187] conducted a study using a plant-produced SARS-CoV-2 spike protein vaccine in hamsters and found that it elicited heterologous immunity without causing any adverse effects. Hager et al. [188] conducted a clinical trial on a recombinant plant-based adjuvanted COVID-19 vaccine and reported its efficacy and safety in human subjects. These studies highlight the safety of plant-based vaccines in preclinical and clinical settings.

In terms of efficacy and immunogenicity, several studies have shown promising results. Panapitakkul et al. [189] investigated the immunogenic response of a plant-produced S1 subunit protein of SARS-CoV-2 in mice and found that it elicited robust immune responses. O’Kennedy et al. [190] evaluated the immunogenicity of a plant-produced SARS-CoV-2 Beta spike VLP vaccine in rabbits and observed significant antibody responses. Mamedov et al. [191] produced and characterized nucleocapsid and RBD cocktail antigens of SARS-CoV-2 in Nicotiana benthamiana plants, demonstrating their potential as vaccine candidates against COVID-19. Additionally, studies on plant-based vaccines for other diseases have also reported positive results. Hodgins et al. [192] conducted a study using a plant-derived virus-like particle influenza vaccine and observed a broad immune response and protection in aged mice. Pillet et al. [193] conducted clinical trials on a plant-derived virus-like particle influenza vaccine in adults and reported immunogenicity and safety. These studies, along with other reports provided in the revised manuscript, encompass a range of preclinical, clinical, and real-world data on the safety, efficacy, and immunogenicity of plant-based vaccines against COVID-19 and other diseases. By incorporating these findings, we aim to provide a comprehensive discussion on the available data to support the potential of plant-based vaccines as a viable immunization strategy.

## 6. Status of Plant-Based Vaccines against COVID-19

The use of plant-based platforms for vaccines has grown significantly during the past few decades. Numerous researchers have attempted to produce vaccines in plants since the first attempt was made by Hiatt et al. [194]; among them, the subunit protein expression of surface antigen of Streptococcus mutants and hepatitis B in tobacco [195] and heat-labile toxin B subunit in potato [196] are noteworthy. However, due to transgenic concerns, the US Food and Drug Administration (FDA) had yet to license any plant-based vaccines as of 2015. Later, the US Department of Agriculture (USDA) approved only two products, the Newcastle disease virus (NDV) vaccine for poultry vaccination [197] and plant-made single-chain fragment variable monoclonal antibody (scFvmAb) for hepatitis B virus (HBV), which is used in the production of a recombinant HBV vaccine for human vaccinations in Cuba [198]. Since then, a wide range of plant-based vaccines have been created to protect against a number of human diseases, including polio [199], human papillomavirus [200], Ebola [201], zika [202], dengue [203], influenza [204], and HIV infection [205]. As of now, several plant and viral vector-based platforms have been used to create over 73 experimental vaccines, including 61 prophylactic and 12 therapeutic vaccines [206]. A lot of vaccine antigens produced by plants are currently undergoing clinical or advanced preclinical testing. Many industrial players, like PlantForm, IBio Inc., Mapp Biopharmaceutical, Inc., Pfizer Inc., Ventria Bioscience, Medicago Inc., Greenovation Biotech GmbH, Kentucky BioProcessing, PhycoBiologics Inc., Synthon, Fraunhofer IME, Healthgen, Planet Biotechnology, and Icon Genetics GmbHthe, are now making investments in the global market for plant-based biologics. With growing demand, it is projected that the global market for plant-based vaccines will be worth 2672.05 million USD in 2028 from 1143.72 million USD in 2021 and rise at a rate of 12.9% from 2021 to 2028 [207]. As a result, the aforementioned data point to a strong trend that the sector for plant-based vaccines will experience significant growth over the coming years.

The production of biologics in plants holds pivotal importance compared to animal and microbial-cell culture-based platforms. Earlier studies have highlighted the advantages of plants as vaccine platforms (molecular pharming) due to their low production costs, high scalability, and increased safety, as they rarely carry human or animal pathogens [208]. Additionally, the lower capital investment required to establish economical greenhouses with suitable crop care facilities is comparable to the expensive facilities, bioreactors, and high-cost culture media needed for cell culture-based vaccine production systems. Furthermore, the introduction of good manufacturing facilities combined with vertical farming, hydroponics, and LED lighting helps to scale up production in limited space at a rapid pace. This platform also exhibits superiority through innovations in expression vectors, transitioning from transgenic to transient expression. Transient expression offers greater flexibility and speed that cannot be matched by other production technologies like mammalian cell culture [209]. The development of ‘deconstructed’ viral vector systems (e.g., magnICON, geminiviral, and pEAQ) has successfully addressed challenges related to insufficient protein expression levels, consistency, and speed of biologic production in plants [210,211,212]. For example, using deconstructed viral vectors based on transient expression, it is possible to achieve the production of up to 5 milligrams of monoclonal antibodies (mAbs) per gram of fresh leaf weight within 2 weeks. Plant-based platforms have been utilized to produce vaccines and therapeutics for influenza viruses, papillomavirus, hepatitis B virus, Ebola viruses, rabies virus, bunyaviruses, flaviviruses, and more [213]. Therefore, plant-based production technologies should be exploited to rapidly develop low-cost vaccines and antibodies for therapy, prophylaxis, and diagnosis against the COVID-19 pandemic.

### 6.1. Production of Subunit Vaccines in Plants

To meet the global demand for billions of doses of the COVID-19 vaccine, the expression of subunit proteins in plants is the best option. The target protein can be expressed in plant cells through either transient expression or transgenic expression. For transgenic expression, the host plant undergoes Agrobacterium-mediated stable transformation. Previous studies targeted the expression of both full-length and truncated versions of the spike (S) protein of SARS-CoV [214]. The development of recombinant protein subunit vaccines against different coronaviruses (CoVs) serves as clear evidence. Initially, the transgenic expression of recombinant (full and truncated) versions of the glycoprotein S from the transmissible gastroenteritis coronavirus (TGEV) of swine in Arabidopsis paved the way for subunit vaccine production in plants against CoVs [215,216]. Later, the S1 glycoprotein of the infectious bronchitis virus (IBV) was selected for expression in transgenic potato, which induced desirable immunogenicity in mice and chickens against CoVs when orally administered, comparable to other commercial vaccines [217,218]. Similarly, the spike protein of the transmissible gastroenteritis virus (TGV) was used to produce an oral vaccine in corn seed, further boosting immunity in swine against CoV [219]. However, very limited experiments have been conducted targeting Human coronaviruses (HCoVs) using plant-based vaccines. Previous studies reported significant accumulation of truncated spike (S) protein of SARS-CoV in the cytosol of nuclear-transformed tobacco and lettuce plants, as well as in the chloroplasts of plastid-transformed tobacco plants and the cytosol of tobacco via transient expression to produce a safe oral recombinant subunit vaccine, although the immunogenicity of that vaccine has not been confirmed [220]. To evaluate the efficacy of plant-based vaccines, the N-terminal fragment of the spike protein (S1) of SARS-CoV was expressed in transgenic tomato and low-nicotine tobacco, and the vaccine showed immunogenic ability in mice when orally administered [221]. Another recombinant vaccine expressed in *N. benthamiana* using the nucleocapsid (N) protein of SARS-CoV generated adequate humoral and cellular immune responses in mice when the plant extract was emulsified in Freund’s adjuvant and administered through intraperitoneal injection [222]. However, none of these vaccines have advanced to further clinical trials, but they have provided a roadmap for future vaccine development.

In the current pandemic situation, the focus has intensified on generating billions of doses of the vaccine using plant-based platforms. Scientific organizations have come forward with their well-established ‘rapid vaccine production’ systems, including good manufacturing facilities such as vertical farming and automated hydroponics [223,224]. For the production of the recombinant subunit vaccine for COVID-19, Beijing CC-Pharming Ltd., Beijing, China in collaboration with iBio Inc., Bryan, TX, USA), started experimentation with commercial confidentiality. The gene encoding the spike protein is possibly inserted into a tobacco mosaic virus-based proprietary expression vector (Beijing CC-Pharming Ltd.), which is then transiently expressed in glycan-engineered N. benthamiana grown in the FastPharming System™ (iBio Inc., New York, NY, USA) to obtain large quantities of high-quality antigen. The FastPharming System™ has previously been used to produce the E2 subunit vaccine for Classical Swine Fever Virus (CSFV) [225]. This plant-based platform offers large-scale rapid production of antigens at significantly lower costs and shorter timeframes (about weeks) compared to cell-based systems [161]. Subsequently, many new initiatives have been undertaken to develop subunit protein-based antigens for SARS-CoV-2. One such recombinant vaccine (subunit proteins) developed by BaiyaPhytopharm Co. Ltd. (Bangkok, Thailand) has reached clinical phase I trials (Table 4).

### 6.2. Production of VLP Vaccines in Plants

The soluble recombinant protein subunits may not always be efficient enough to induce an adequate immune response against the virus pathogen. To avoid such problems, scientists are now focusing on assembling individual proteins into defined particles, known as pseudoparticles, nanoparticles, or VLPs. These particles display various epitopes on their surface, allowing them to interact differently with immune cells and trigger both humoral and cellular immune responses. The generation of VLPs has been prioritized with the development of various virus-based vector systems, such as TMV and Cowpea mosaic virus (CPMV) [206,226]. One such VLP platform, known as Proficia^®^/VLPExpress™, consists of an Agrobacterial binary vector/CPMV-HT vector-based transient expression system developed by Medicago Inc. (Québec, Canada) for rapid production of VLPs and therapeutic proteins in *N. benthamiana* [227]. It has immense capacity for vaccine production, such as producing over 10 million doses of vaccine against H1N1 influenza in 1 month [228], and now it is expected to produce up to 20 million and 100 million annual doses for Quebec and North Carolina, respectively [229]. Initially, this platform was used for the production of VLPs against Influenza A viruses (H1N1, H5N1). Such VLP vaccines elicited a broad immune response in mice [192] and underwent clinical trials phase II (gov. NCT00984945) [193]. Now, Medicago Inc. is utilizing the same platform for the rapid manufacturing of VLP-based vaccines displaying the spike protein of SARS-CoV-2. Although the specific information on the VLP fabrication strategy remains the company’s trade secret, VLPs resembling coronavirus particles can be created within 20 days after obtaining the target gene of SARS-CoV-2 [227]. In this way, Medicago Inc. has taken the first step towards vaccine development against COVID-19, and in preclinical trials, the candidate vaccine induces an immune response in mice within 10 days after a single application [229]. It has undergone different phases (I, II, III and IV) of clinical testing, showing higher immunogenicity by generating neutralizing antibodies (NAb) and inducing spike protein-specific interferon-γ and interleukin-4 cellular responses [230], and has finally been approved in Canada with the commercial name COVIFENZ^®^. This positive response boosts expectations for many more clinical trials soon.

Similarly, in collaboration with the Infectious Disease Research Institute (IDRI), iBio, Inc. (USA) is engaged in developing a VLP-based vaccine to counter the COVID-19 outbreak. For constructing VLPs, the receptor-binding motif in the S protein of SARS-CoV-2, an important ‘antigen’-encoding gene [231,232], is fused with self-assembling protein partners, namely, the core proteins of human Hepatitis B virus (HBV) and Woodchuck hepatitis virus (WHV) [233], and then expressed in tobacco plants using their proprietary Fast Pharming system [234] to generate the VLPs. After purification, VLP structures are characterized as nanoparticles with a diameter of 40–50 nm, displaying desirable antigens [235]. Such VLPs are also decorated with oligomannose molecules that resemble the architecture of naturally occurring viruses. The purified VLPs are further processed by mixing them with different novel adjuvants to maximize uptake by antigen-presenting cells and trigger both humoral and cellular immune responses. These VLP-based vaccines are designated as IBIO-200 and IBIO-202, representing promising vaccine candidates for managing COVID-19 [234,236]. Furthermore, the FastPharming Facility™ enables fast scalability of millions of vaccine doses, raising expectations for the commercial availability of the IBIO202 vaccine to the public. VLPs synthesized in the plant system provide additional safety for humans, with the opportunity for massive scale-up in production through plant molecular farming [235]. Many other companies and research institutes have also announced the production of VLPs embedded with the spike protein of SARS-CoV-2 in plant-based transient expression systems for managing COVID-19 (Table 4). However, only a few of them have reached the clinical trial phase. Although the progress of plant-produced vaccines is slow, the promising results of COVIFENZ^®^ and other plant-derived candidate vaccines against influenza and Zika virus in recent years encourage the expectation of the commercialization of many more plant-derived vaccines in the upcoming decade.

**Table 4 vaccines-11-01347-t004:** Development of plant-based vaccine against coronaviruses.

Type of Vaccine	Name of Vaccine	Epitope	Production Strategy *	Formulation	Administration	Immune Response	Developer	Development Phase	Ref.
Recombinant protein subunit	Subunit protein	Spike (S) protein	iBio’s Fast-Pharming facility (VLPExpress™)	-	-	-	iBio Inc. (Bryan, TX, USA) and CC-Pharming Ltd. (Beijing, China)	Pre-Clinical	[234]
	IBIO-201	Spike (S)protein	-	Formulated with LicKMTM adjuvant	Intramuscular and intranasal	Immune response to SARS-CoV-2	iBio Inc. (Bryan, TX, USA)	Pre-Clinical	[236,237]
	Baiya SARS-CoV-2 Vax 1	-	-	Formulated with alum adjuvant	-	Immune responsein mice and monkeys after two doses	Baiya Phytopharm Co., Ltd. (Bangkok, Thailand)	Clinical trial (Phase I)	[238]
	Subunit protein	Antigen cocktail (S, N, N+RBD protein)	-	Antigen cocktailinAlhydrogel adjuvant	IM immunization at0 and 21days	Elicitation of antibodies in mice	Akdeniz University, (Antalya, Turkey)	Pre-clinical	[191]
	Subunit protein	S-protein	NXS/T Generation™ platform	glycan-enhanced (GE) Spike protein	Intranasal delivery	Broad spectrum immunity in hamsters against SARS-CoV-2 variants	University of Cape Town, (Cape Town, South Africa)	Pre-clinical	[187]
	Recombinant Protein	S1 protein with the Fc domain of human immunoglobulin G1 (IgG1)	Geminiviral expression vector	S1-Fc protein formulated withaluminum hydroxide gel(adjuvant)	IM injection at0 and 21 days	RBD specific antibodies and T cell immune responses in mice	Chulalongkorn University (Bangkok, Thailand)	Pre-clinical	[189]
	Recombinant vaccine	Receptor-binding domain	-	-	-	Formation of neutralizing antibodies against Wuhan strain	G+Plus Life Science Co., Ltd. (Seoul, Republic of Korea)	Pre-clinical	[239]
Virus like particle (VLPs)	Covifenz	S protein	Proficia^®^ Technology	Recombinant VLP adjuvanted withAS03	IM injection with 2 dosesin 21 days apart	Interferon gamma (IFN-γ) and IL-4 responses	Medicago Inc., (Quebec, Canada) in collaboration with Glaxo-Smith Kline (Middlesex, United Kingdom)	Approved in canada; Clinical trial in Argentina, Brazil, U.K.Ireland, USA, Japan	[188]
	IBIO-200 (glycosylated/non-glycosylatedVLPs)	S protein	FastPharming System	Special adjuvant (of IDRI)	-	Specific immunity in mice	iBio, Inc. (Bryan, TX, USA) & Infectious Disease Research Institute (Seattle, WA, USA)	Pre-Clinical	[234]
	IBIO-202	N protein	FastPharming System	Special adjuvant	Intramuscular and Intranasal	broader range defense against SARS-CoV2variants	iBio Inc. (Bryan, TX, USA)	Pre-Clinical	[237]
	KBP 201	RBD of S protein fused to a human IgG1 Fc domain	Tobacco mosaic virus (TMV)-based expression systm	Chimeric VLPsconjugated with CpG 1018 (adjuvant)	Intramuscular injection at an interval of 21 days	Specific immunity in pre-clinical trials	Kentucky Bioprocessing Inc. (Owensboro, KY, USA)	Clinicaltrial (Phase II)	[240]
	VLPcandidates displaying the S-protein	spike (S) surface glycoprotein	-	VLPs mixed with three adjuvants i.e., oil-in-water, synthetic oligodeoxynucleotide (ODN)	Two doses at 0 and 21 days	Elicitation of antibodies to cross-neutralise Delta (B.1.617.2) and Omicron variant in white rabbits	Council for Scientific and Industrial Research, (Pretoria, South Africa)	Pre-clinicaltrial	[190]

* Transient expression in *Nicotiana benthamiana* (N.B.).

## 7. Limitations of Plant-Based Vaccine Production

With regard to plant-based vaccinations, there are a number of possible drawbacks and issues. Most notably, the technical clearance for commercial manufacturing and consumption is constrained by the development of plant-made vaccines utilizing a transgenic technique. Plant-based vaccines, like other biopharmaceuticals, should be devoid of contaminants, such as transgenes and antibiotic resistance marker gene products, which must all be assessed according to the same standards [241]. Transiently transfected plants and plant cell cultures can be used to rule out transgene expression. Agroinfiltration or plant viral vectors may be used in numerous transient expression methods, which call for sophisticated regulation that applies to all plant transgenic technologies. For creating plant-based vaccines for clinical use in humans and animals, good manufacturing practice (GMP) compliance should be established [242]. Governmental regulatory guidelines must be followed to secure public acceptance. Additionally, there are still a number of significant risks and concerns connected to plant-based biologics. The issue with oral tolerance and allergenicity is the most serious [243]. The post-translational changes after N-glycosylation may cause hypersensitivity reactions to other proteins, thus compromising the bioefficacy of many plant-based vaccines [244]. Plant glyco-engineering would be best option to avoid such undesirable post-translational modification [245]. The risks and limitations associated with plant-based vaccines must be addressed to bring them into the public domain.

## 8. Path Ahead for the Improvement of Plant-Based Platform

Since the commercialization of ZMapp™, a cocktail of three monoclonal antibodies (mAbs) for managing the Ebola virus, plant-made biologics have gained popularity, and many more are ready to enter the public domain. However, these have not yet been widely adopted systematically worldwide. Therefore, there is a need for further refinement in vaccine research with the aim of increasing production velocity, diversifying plant-based platforms, enhancing regulatory flexibility, and facilitating wide-scale applicability. The production of plant-based vaccines relies heavily on the availability of suitable expression platforms. Transitioning from transgenic to transient expression systems not only ensures rapid production but also enhances the quality and quantity of vaccines within a short time span [171]. This is essential for the expedited production of billions of vaccine doses due to the rapid speed and spread of pathogens during pandemics. Innovations and cutting-edge technologies have led to the reorganization of plant virus genomes (such as CMV, CPMV) into suitable expression vectors that are highly efficient in designing self-assembled VLPs and displaying multiple epitopes in a large number of plant hosts [206]. Furthermore, the exploration and utilization of plant virus genomes are necessary to accelerate the velocity and magnitude of vaccine production.Plant viruses have emerged as valuable tools for the expression of foreign proteins. By leveraging the natural infection machinery of plant viruses, scientists can engineer them to deliver and express desired proteins in plants. Plant virus-based expression systems [246,247,248,249] offer numerous advantages, such as high expression levels, rapid production, scalability, safety, and cost effectiveness. These systems have been successfully employed to produce a wide range of proteins.

Multiplicity in plant-based production can also be encouraged through the introduction of appropriate plant platforms. So far, different tobacco plants (*Nicotiana* sp.) have been extensively used as biofactories for biologics [158], but they are not suitable for direct consumption as food or feed and therefore not accepted in food or feed chains. Additionally, the presence of toxic alkaloids and phenolics in many tobacco cultivars necessitates prolonged purification before application. To address these issues, alternative leafy vegetables such as alfalfa, lettuce, spinach, or fruits like banana or cucurbits can be favored as suitable alternatives for plant molecular pharming [250]. These alternatives would be preferable for low-cost vaccine production achieved through direct oral consumption. This not only reduces purification costs but also improves vaccine yield by preventing losses during purification. The vaccines can be delivered in the form of gelatin pills or tablets encapsulating freeze-dried plant biomass [251]. The main advantage lies in the elimination of the painful administration of injectable vaccines, thus avoiding the requirement for medical staff and trained personnel, which are crucial for achieving wide vaccination coverage in developing countries.

Furthermore, regulatory agencies must adapt their regulations based on requirements and technological advancements. Considering the safety and efficacy of plant-made vaccines against influenza and Ebola, regulatory agencies should be flexible in conducting clinical trials in humans [252]. This flexibility will not only encourage industries engaged in plant-made biopharmaceutical production but also reduce the time gap between production incubation and large-scale distribution. Additionally, there is a demand for more public–private collaborations to facilitate the growth of industries producing plant-based biologics. As part of pandemic preparedness, financial, social, and regulatory support from governments is imperative to make this venture more successful. If all these factors come together, plant-based vaccines can meet global demands, especially during disease outbreaks.

## 9. Conclusions

In the race to tackle COVID-19, scientists worldwide are searching for suitable solutions using every possible option. In such a situation, the pharmaceutical industry that produces drugs and vaccines offers a glimmer of hope in the midst of the pandemic’s darkness, as the world remains unprepared for this novel outbreak. The ineffectiveness of existing drugs and vaccines against coronavirus infections further raises concerns. Therefore, the discovery of promising vaccines becomes the only choice. With the outbreak of COVID-19, the employment of innovative approaches like artificial intelligence facilitates the production of viable vaccines by rapidly identifying epitopes and their effectiveness in generating a broad-spectrum immune response against SARS-CoV-2. Producing such vaccines using a plant-based platform, in compliance with good manufacturing practices such as vertical farming, hydroponics facilities, and intensive lighting, promotes a higher yield of vaccines in a shorter time. Not only are these plant-made vaccines faster and cheaper, but they are also safer than other vaccines. They are also highly immunogenic and can be easily delivered to everyone through oral administration. Thus, plant-based vaccines would be a novel choice to reach billions of people worldwide during a pandemic crisis.

## Figures and Tables

**Figure 1 vaccines-11-01347-f001:**
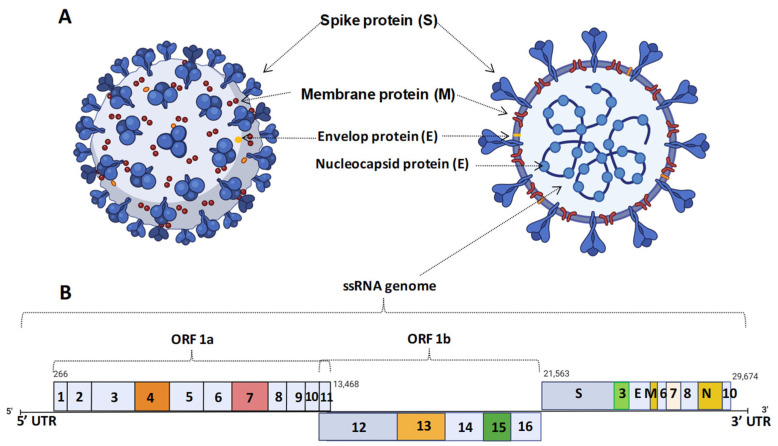
Architecture and genome organization SARS-CoV-2. (**A**) SARS-CoV-2 has a spherical structure with 80–90 nm diameter, which is composed of a nucleocapsid formed by condensation of nucleocapsid (N) protein with RNA genome and membrane associated (M) protein and surrounded with an outer lipid envelope (E), projecting numerousspike (S) glycoproteins. (**B**) The SARS-CoV-2 has single stranded(ss), positive sense RNA genome of approximately 29.9 kb length and is comprised of two large genes (ORF1a and ORF1b) at the 5′ end encoding 16 nonstructural proteins, viz., nsp1–nsp10 from ORF1a and nsp1–nsp16 from ORF1b to from a replication–transcription complex (RTC), and eight genes at the 3′end encoding four structural proteins, viz., spike protein (S), envelope (E) protein, membrane (M) protein, and nucleocapsid (N) protein along with other accessory proteins.

**Figure 2 vaccines-11-01347-f002:**
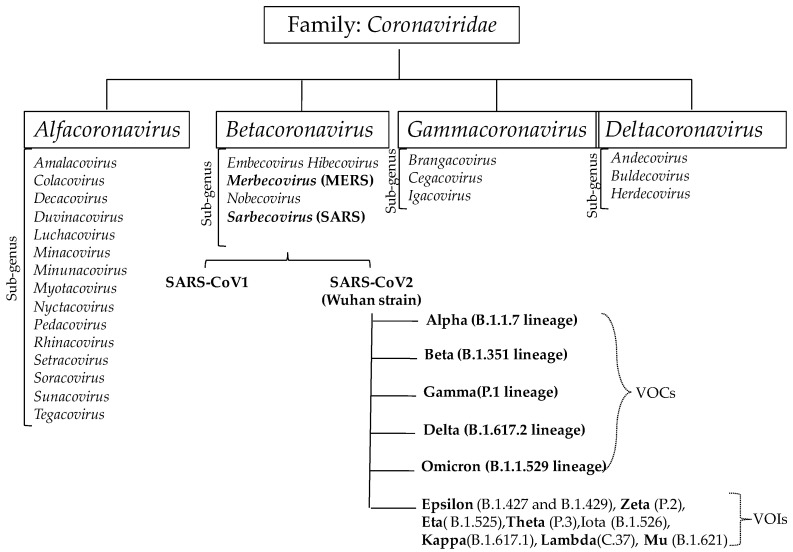
A schematic diagram representing the classification of different coronaviruses with special reference to SARS-CoV-2 with their strains. Coronaviruses belong to the family *Coronaviridae*, under the order *Nidovirales*. They are classified into four genera (alfa, beta, gamma, and delta), and each genus can be further divided into different subgenus. Severe acute respiratory syndrome coronaviruses (SARS-CoVs) belong to the genus *Betacoronavirus* along with Middle East respiratory syndrome coronavirus (MERS-CoV). SARS-CoV-2 further rapidly evolved into different strains through mutation from its original Wuhan strain.

**Figure 3 vaccines-11-01347-f003:**
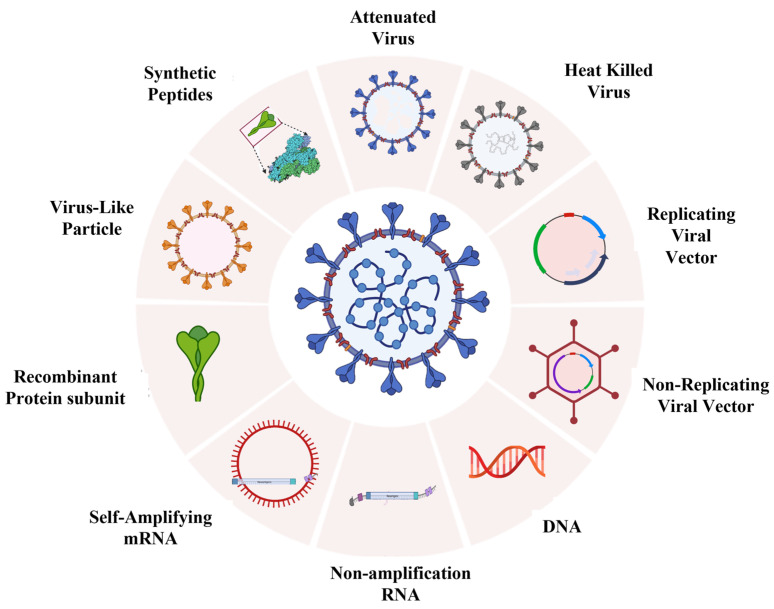
Different types of vaccines produced for COVID-19.The whole-cell (pathogen)-based vaccines include live-attenuated and heat-killed vaccines that are developed by inactivating the pathogen to lose its infectivity while maintaining immunogenicity. They mimic the structure of the pathogen without causing disease. Replicating and non-replicating vector-based vaccines are designed by expressing the full-length spike (S) protein of SARS-CoV-2 through adenovirus-based vectors.In the nucleic acid-based vaccine category, DNA and mRNA-based vaccines are the most prominent. After delivery, DNA vaccines have to enter the cell nucleus to transcribe into mRNA, which then moves back to the cytoplasm and uses the host protein translation system to generate target antigens. On the other hand, self-amplifying and non-amplifying mRNA vaccines are delivered directly into the cytoplasm to express target antigens. Protein subunit vaccines are expressed in different platforms such as bacteria, fungi, insects, and plants through genetic engineering of the pathogen. They are purified and applied with adjuvant to induce immunity. VLPs are generated by assembling the structural proteins in a manner that mimics the structure of the pathogen without causing disease while retaining immunogenicity.

**Figure 4 vaccines-11-01347-f004:**
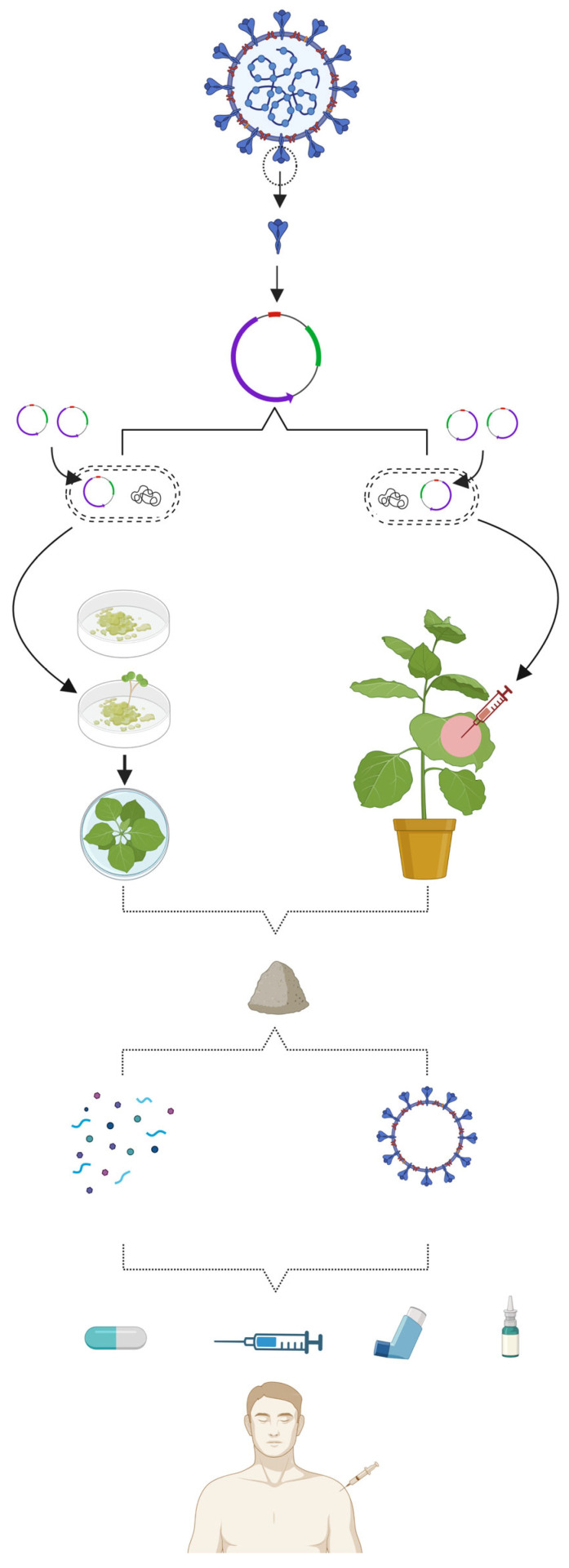
Expression strategies of vaccines in plant system. The spike protein sequence of SARS-CoV-2 is integrated into a binary construct, such as pCAMBIA 2300 and delivered into the plant using either agro-transformation or agro-infiltration techniques. Transgenic transformation of *Nicotiana benthamiana* results in the expression of subunit proteins only. However, transient expression within *Nicotiana benthamiana* allows for the generation of both subunit proteins and VLP as vaccine candidates. These vaccines can be administered via oral, nasal, or injection-based delivery routes.

**Table 1 vaccines-11-01347-t001:** List of vaccines approved by WHO for COVID-19.

Sr No.	Candidate Vaccine	Content	Approved in Countries	Administration	Developer/Manufacturer
1	CoronaVac	Inactivated virus (Vero cell)	56	Intramuscular injection of 2 doses	Sinovac Life Sciences Co., Ltd. (Beijing, China)
2	Covilo	Inactivated virus (Vero cell)	93	Intramuscular injection of 1, 2 or 5 doses	Beijing Institute of Biological Products Co., Ltd. (BIBP) (Sinopharm, Beijing, China)
3	COVAXIN	Inactivated virus (Whole Virion)	14	Intramuscular injection of 1, 2, 5 or 10 doses	Bharat Biotech International Ltd. (Hyderabad, India)
4	Covishield	Non Replicating Viral Vector ChAdOx1-S (recombinant)	49	Intramuscular injection of 2 or 10 doses	Serum Institute of India Pvt. Ltd. (Pune, India)
5	VAXZEVRIA	Non Replicating Viral Vector ChAdOx1-S (recombinant)	149	Intramuscular injection of 2 or 10 doses	Oxford/AstraZeneca with SK Bioscience Co. Ltd. (Gyeonggi-do, Republic of Korea)
6	Convidecia	Non Replicating Viral Vector Ad5-nCoV-S [Recombinant])	10	Intramuscular injection of 1 and 3 doses (0.5 mL)	CanSino Biologics Inc. (Tianjin, China)
7	Jcovden	Non Replicating Viral Vector Ad26.COV2-S (recombinant)	113	Intramuscular injection of 5 doses	Janssen (Johnson & Johnson) (Beerse, Belgium)
8	Comirnaty	mRNA vaccine (nucleoside modified)	149	Intramuscular injection of 2 or 6 doses (30 µg, 0.3 mL each)	Pfizer/BioNTech (Goldgrubae, Germany)
9	SPIKEVAX	mRNA(1273)Vaccine (nucleoside modified)	88	Intramuscular injection of Intramuscular injection of 10 doses (0.5 mL per dose)	Moderna Biotech (Madrid, Spain)
10	Covovax	SARS-CoV-2 rS Protein Nanoparticle (Recombinant)	6	Intramuscular injection of 1 dose and 10 doses (0.5 mL per dose)	Serum Institute of India Pvt. Ltd. (Pune, India)
11	Nuvaxovid	(SARS-CoV-2 rS [Recombinant, adjuvanted])	40	10 doses (0.5 mL per dose)	Novavax CZ a.s. (Jevany, Czech Republic)

Source: https://covid19.trackvaccines.org/agency/who/ (Last updated 2 December 2022).

**Table 2 vaccines-11-01347-t002:** Different types of vaccines with pros and cons.

Class	Types	Description	Application	Pros	Cons
Pathogen-based vaccine	Inactivated pathogen	Whole celled pathogens killed by chemical, heat or radiationtreatment	Difficulty in epitope designing for highly mutating pathogens like influenza, polio, etc	1. Easy to develop 2. Immune response induced by original pathogen	With emergence of new strains, the immunogenicity of vaccines reduced
	Live-attenuated pathogen	Genetically engineered, weakened or attenuated strains of pathogen with reduced virulence	Established platform for multiple human pathogens, viz., measles, mumps, rubella, chicken-pox, etc	Induce strong cellular and humoralimmune responses	1. Lengthy andtime-consuming development process 2. Risk of virulence reversion of virus strain via recombination and mutation
Recombinant vector vaccine	Replicating Virus Vector	Efficient expressionof antigen using replicating virus vector-based expression system	Efficient delivery of antigen into human cells and tissues	Induction of cellular and humoral immune response at low dose	1. Pre-existing immunity against the virusvector reduced replicability of the construct, 2. Safety and immunogenicity issues
	Non-replicating virus vectors	Efficient expressionof antigen using non-replicating virus vector-based expression system	Efficient delivery of antigen into human cells and tissues	Induction of cellular and humoral immune response at low dose	1. Risk of reversion of pathogenicity is very minimal 2. very safe to use
Nucleic acid-based vaccine	DNA	Antigen encoding DNA (gene sequence) is delivered to human cell using bacterial plasmid	Efficient delivery of antigen into human cells and tissues	Easy to design, Rapid manufacture, Noninfectious nature	1. Very difficult to deliver into human cell, may require some special care; 2. Low transfection and lesser protein expression
	RNA	Antigen expression through through self-replicating or non-replicating mRNA	Efficient administration of mRNA via lipid-based delivery systems like lipoplexes and polyplexes	Noninfectious molecules induce humoral and cellular immune responses	Risk of side effects like cardiac arrest
Protein-based vaccine	Protein subunit	Heterologous expression of certain part of the pathogenshowing immunogenicity used as protein subunit vaccine	Preparation of Subunit vaccine formulations by mixing purified antigens with potent adjuvants	Low risk, Safe and stable, fast manufacturing,	Lower immunogenicity; Requirement of adjuvant or conjugate to increase immunogenicity
	VLP	Aggregation of protein forming a virus like configuration without anyvirus genome	Self-assembly of virus capsid protein forming a nanoparticle like structure with potential antigenic response; applicable against different diseases	Noninfectious, Strong humoral response, safe for immune compromisedindividuals, More stable than subunit vaccine	Noninfectious, Difficulty in scaling up of production

**Table 3 vaccines-11-01347-t003:** Comparison of different vaccines production platforms.

Parameters	Bacteria (*E. coli*)	Fungi (Yeast)	Insect Cell	Mammalian Cell	Plant
**Amount of Protein expression**	High	Low	Low	Low	High
**Speed of Protein expression**	Very high	High	High	High	High
**Cost**	Very low	Low	Low	High	Medium
**Scalability**	High	High	Medium	Medium	Medium
**Yield**	High	Medium	Medium	Medium	Medium
**Post-translational modification** **(N-glycosylation)**	No	Yes	Yes	Yes	Yes
**Protein purification**	Difficult	Difficult	Difficult	Difficult	Easy *
**Chance of contamination**	Yes	Yes	Yes	Yes	No
**Immunogenicity**	Low	Medium	Medium	Very high	High
**Oral delivery**	No	No	No	No	Yes
**Cool chain transportation**	Require	Require	Require	Require	No

* Using affinity tags.

## Data Availability

Not applicable.

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
