# Peer review of "Exigency of Plant-Based Vaccine against COVID-19 Emergence as Pandemic Preparedness"

_vaccines, 2023, doi:10.3390/vaccines11081347_

Round 1

Reviewer 1 Report

This Review paper addresses a timely and important topic in the context of the COVID-19 pandemic. With additional details, comparisons, and clarification, this review could provide valuable insights into the potential of plant-based vaccines to improve global vaccine access and preparedness for future pandemics.

I have the following recommendations to further improve this manuscript:

 1-    The paper provides a comprehensive overview of the current status of COVID-19 vaccines, challenges in distribution, and the potential benefits of plant-based vaccines. However, it might be helpful to streamline the focus of the paper by emphasizing plant-based vaccines earlier in the text.

 2-    The rationale for the development of plant-based vaccines in the context of COVID-19 is well-presented, but the review could benefit from a more detailed discussion of the underlying technology and mechanisms involved in the production of these vaccines. This would provide readers with a better understanding of how plant-based vaccines differ from other vaccine platforms.

 3-    The authors should elaborate on the advantages and disadvantages of different types of vaccines developed against SARS-CoV-2, as mentioned in the paper. A comparative analysis of the effectiveness, safety, scalability, and cost of each vaccine type would be valuable.

 4-    The review should include a more extensive discussion of the available data on the safety, efficacy, and immunogenicity of plant-based vaccines against COVID-19 and other diseases. This could include results from preclinical studies, clinical trials, and real-world data, if available.

 5-    The authors mention the advantages of plant-based vaccines in terms of manufacturing and distribution but should provide more details on these aspects. For instance, how do the production times and costs compare to those of other vaccine platforms? What are the specific challenges and potential solutions for the cold chain requirements of plant-based vaccines?

 6-    The authors should address potential limitations and concerns associated with plant-based vaccines, such as regulatory hurdles, public acceptance, and the possibility of plant-based allergens in the final product.

 7-    There are some grammatical errors and sentence structure issues throughout the paper that should be addressed for clarity and readability.

 8-    Some abbreviations are used without being introduced first (e.g., VLP). Please introduce abbreviations before using them.

 9-    The authors should ensure that all relevant and recent literature on plant-based vaccines against COVID-19 is cited and discussed appropriately.

 There are some grammatical errors and sentence structure issues throughout the paper that should be addressed for clarity and readability.

Reviewer 2 Report

Authors have collected adequate theoretical information on the status and challenges on vaccines against COVID-19. I appreciate the time and effort that authors have put into this work. However, after careful consideration, I regret to inform that the study could not reach the acceptance of publication in the present form. Especially, the study design is insufficient, there is a lack of diagrammatic and comparative analysis, and the information is not well-arranged. The whole manuscript needs to be redesigned and rewritten. The descriptive information should be represented with appropriate schematic diagrams, and tables with comparative information in a concise way. Some of my viewpoints are following below:

1. The authors are describing coronavirus structure in the section 2, page 2, lines 81 to 86, it could be an appropriate indication to include a coronavirus schematic diagram. Further, authors describing the classification of coronavirus, lines 88 to 99 and lines 147 to 150, it should also include a schematic diagram to represent the classification of different coronavirus.

2. The author is describing about different vaccines in section 4, page 5, line 222, it could be contained a comparative table describing the vaccines and their brief comparison.

3. In section 5, page 11, line 494, it is being discussed “plant-based platform for vaccine production”. However, it should be provided the clear explanation of what is a plant-based vaccine and include a diagram to quickly understand the technique.

4. Table 2. Which method or technique was used to define the parameters like high or low?

5. Some information is repeating. For example, page 12, lines 525 to 531, the challenges were discussed above in the manuscript. Section 6, page 13, lines 568 to 578, the paragraph sounds repeating. The whole paragraph could be removed as it should be discussed directly about the status of plant-based vaccines.

6. Table 3 should be well formatted.

7. Figure 1 and figure 2 are in weak resolution. It should be improved in appropriate resolution (e.g., 600 dpi).

8. References need to be arranged according to the journal's guidelines.

Reviewer 3 Report

A good and comprehensive review.

A few minor issues.

Line 60,61 the meaning is unclear

Line 119. Reference by Wu in 2020 has probably been questioned in more recent studies.

The part of the review up to line 177 looks only to cover references up to 2021 and I suspect the detail needs updating by including references up to 2023.

Round 2

Author Response

We sincerely appreciate the reviewer for providing their valuable feedback. We agree with almost all of their points, and we have incorporated numerous new references to bolster our research. We are confident that these additions will improve the paper's overall quality. In the revised manuscript, we have included all the required information, which is indicated with red highlighting for easy identification. We have addressed the reviewer's queries and concerns in the following section: 

  1. Comments to authors: Several paragraphs in the manuscript are without references. A few examples are given below but these are not limited, a careful revision is required throughout the manuscript. Response: Extensive revision is done and many new references are cited.
  2. Comments to authors: Line 78, “different platforms” Which platform? Give examples of platforms. Response: different platforms for vaccine development, viz., egg-based, prokaryotic, and eukaryotic cell-culture-based systems’ are exemplified
  3. Comments to authors: Figure 1, the caption is short, give different panel names and describe such as A, B and C. Response: The caption is rewritten in two sections (A) and (B)
  4. Comments to authors: There are several typo issues in the manuscript such as “gainingD614G” line 171, “mutant was” line 172,
    there is no space in between the words. Line 130 “Unhale et al., 2020)”. Response: Many typo issues are corrected in this revised version
  5. Comments to authors: Several places the word “etc.” has been written such as line 237, line 309, line 318. It is not recommended to mention “etc.” in the sentences. Response:  Thank you for your feedback. We have carefully reviewed the manuscript and made the necessary changes by removing the use of "etc." in the entered MS, as suggested by the reviewer.
  6. Comments to authors: Page 2, line 53, “affected (98).” It is not clear what is 98, does it represent a reference? Response: It is a reference and the reference number is corrected as [2]
  7. Comments to authors: Page 5, lines 216 to 219, “Whereas, many newly emerging strains,viz.,Epsilon (B.1.427, and B.1.429); Zeta (P.2); Eta( B.1.525); Theta (P.3); Iota (B.1.526); Kappa(B.1.617.1); Lambda(C.37),and Mu (B.1.621) are considered as the variants of interest (VOIs) (Figure 2) keeping the possible risk associated with them(32).” The sentence is not clear and not well formatted. Response: Changed to Whereas, numerous newly evolved strains, including Epsilon (B.1.427 and B.1.429), Zeta (P.2), Eta (B.1.525), Theta (P.3), Iota (B.1.526), Kappa (B.1.617.1), Lambda (C.37), and Mu (B.1.621), are regarded as variations of interest (VOIs) (Figure 2) because to the potential risk, they may pose [47].
  8. Comments to authors: “Table 1. List of vaccines approved by WHO for COVID-19.”. A certain date needs to be mentioned, the latest date of these information of vaccines updated. Response: Last updated 2 Dec., 2022)
  9. Comments to authors: “Manuscript includes some classification information. This information needs to be illustrated in a figure or in a table. For instance, section 4.1, 4.2, 4.3, 4.4 and their subsections are describing different types of vaccines including “Pathogen based vaccine, Recombinant vector vaccine, Nucleic acid-based vaccine, Protein-based vaccine”. This information needs to be presented in a schematic diagram to signify the class and their subcategories. It could be something like that. Replace table 2 and figure 3 with a schematic diagram. Response: 

    Thank you for your feedback and suggestions. After careful consideration and discussions within our team, we have decided to retain both Table 2 and Figure 3 in the manuscript. Table 2 provides a detailed and comprehensive comparative analysis of different types of vaccines, including their advantages and disadvantages. This tabular format allows readers to quickly grasp the information and make informed comparisons.

    Additionally, removing Table No. 2 poses challenges, as Reviewer I has already suggested including a comparative study of different types of vaccines with their respective advantages and disadvantages.

    On the other hand, Figure 3 complements the table by providing a visual representation of the classification information from sections 4.1, 4.2, 4.3, and 4.4. It helps readers to better understand the relationships between different vaccine categories and their respective subcategories.

    By retaining both Table 2 and Figure 3, we believe that we can present the information in a clear and comprehensive manner, benefiting the readers and improving the overall quality of the manuscript.

  10. Comments to authors: Table 4, text and format could be improved to arrange the content in the table properly. Response: As per the reviewer's suggestion, we have formatted Table 4 and revised the text to provide a more structured and coherent representation of the data. The changes have significantly improved the overall clarity and presentation of the manuscript.

  11. Comments to authors: References are still not formatted and not according to the journal’s guidelines. Several typo issues are found such that some journal names are abbreviated, and others are not. Also, references are not recommended to include “[CrossRef] [PubMed] or DOIs” etc. Response: References are formatted according to the journal’s guidelines and “[CrossRef] [PubMed] or DOIs” etc.” are removed.